

# Hierarchical creep cavity formation in an ultramylonite and implications for phase mixing

James Gilgannon[1,2], Florian Fusseis[2], Luca Menegon[3], Klaus Regenauer-Lieb[4], and Jim Buckman[5]

[1]Institute of Geological Sciences, University of Bern, Baltzerstrasse 1+3, CH-3012 Bern, Switzerland
[2]School of Geosciences, The University of Edinburgh, Grant Institute, Edinburgh EH9 3JW, UK
[3]School of Geography, Earth and Environmental Sciences, Plymouth University, Plymouth PL4 8AA, UK
[4]School of Petroleum Engineering, The University of New South Wales, Kensington NSW 2033, Australia
[5]Institute of Petroleum Engineering, Heriot-Watt University, Edinburgh EH14 4AS, UK

*Correspondence to:* J. Gilgannon (james.gilgannon@geo.unibe.ch)

**Abstract.** The dispersal of monomineralic quartz domains in a quartzofeldspathic ultramylonite is interpreted to be the result of the emergence of syn-kinematic pores, called creep cavities. The cavities can be considered the product of two distinct mechanisms that formed hierarchically: Zener-Stroh cracking and viscous grain boundary sliding. In initially thick and coherent quartz ribbons deforming by grain size-insensitive creep, cavities were generated by the Zener-Stroh mechanism on grain-

boundaries aligned with the YZ plane of finite strain. The opening of creep cavities promoted the ingress of fluids to sites of low stress. The local addition of a fluid lowered the adhesion and cohesion of grain-boundaries and promoted viscous grain boundary sliding. With the increased contribution of viscous grain boundary sliding, a second population of cavities formed to accommodate strain incompatibilities. Ultimately, the emergence of creep cavities is interpreted to be responsible for the transition of quartz domains from a grain size-insensitive, to a grain size-sensitive rheology.

**1   Introduction**

Microstructural observations of shear zones in nature and experimental investigations of monomineralic systems in the laboratory have demonstrated that the evolution of a ductile fault rock through the mylonite series can entail a switch from a dislocation creep-controlled (grain size-insensitive (GSI)) to a diffusion creep-controlled (grain size-sensitive (GSS)) bulk rheology (e.g. Etheridge and Wilkie, 1979; Poirier, 1980; Kilian et al., 2011). In quartzofeldspathic rocks at mid-crustal condtions,

this progressive evolution often leads to the development of distinct microstructural elements in close spatiotemporal proximity: feldspathic porphyroclasts, monomineralic quartz bands and well mixed, fine-grained, polyphase domains (fig. 1). Each of these elements have been shown to accommodate deformation differently, e.g. feldspars fracture and react, quartz experiences GSI creep while the polyphase domains deforms by GSS processes (Mitra, 1978; Kerrich et al., 1980; Behrmann and Mainprice, 1987; Fliervoet et al., 1997; Stipp et al., 2002; Tullis, 2002). Generally speaking, with ongoing deformation the

proportion of fine-grained material deforming by GSS creep increases syn-kinematically, so that the polyphase domains form an interconnected weak layering. It is the establishment of these well mixed, anti-clustered polyphase domains that is recog-




nised to ultimately promote a switch in the bulk rheology of the rock (Handy, 1994; Herwegh et al., 2014). However, the exact modes by which these domains are established are still poorly understood and the subject of intense research.

One of the details under investigation is the role of fluids and their pathways (for contrasting models see: Etheridge et al., 1984; Paterson, 1995; Fusseis et al., 2009). Fusseis et al. (2009) postulated that in ultramylonites, fine-grained polyphase do-
5 mains deforming by viscous grain boundary sliding (VGBS) develop a *dynamic granular fluid pump*. This pump operates during deformation and utilizes a syn-kinematic porosity known as creep cavitation. Here, creep cavitation is defined as a porosity that results from entropy production during deformation. A stringent consequence of this definition is that pore nucleation must arise directly out of the active deformation mechanism's response to the shortening and stretching of the rock mass. It is clear that a model generating fluid pathways in the middle crust that does not require brittle fracturing has signifi-
10 cant implications for the controls of the exchange of fluids between the hydrostatic and lithostatic pore fluid pressure regimes (Ingebritsen and Manning, 2010), phase nucleation in mylonites (Kruse and Stünitz, 1999) and by extention, rheology.

The phenomenon of creep cavitation has been well described in material science, with several distinct types of creep cavitation being distinguished (Riedel, 1987). To date, geological research has identified evidence for creep cavitation in natural ultramylonites from the middle crust (Behrmann and Mainprice, 1987; Mancktelow et al., 1998; Herwegh and Jenni, 2001;
Fusseis et al., 2009; Kilian et al., 2011; Rogowitz et al., 2016) and the lower crust (Závada et al., 2007; Menegon et al., 2015), as well as in xenoliths (Rovetta et al., 1986). Experimental work has shown that octachloropropane, quartzite, diabase, feldspar aggregates, anorthite and diopside aggregates, and olivine and clinopyroxene aggregates can develop creep cavities (Caristan, 1982; Hirth and Tullis, 1989; Ree, 1994; Dimanov et al., 2007; Rybacki et al., 2008; Précigout and Stünitz, 2016). This data set is small but diverse and suggests that creep cavities can occur in many types of deforming rocks across varying pressure,
temperature and rate conditions.

The wide variety of metamorphic conditions at which cavities form suggests that a range of micro-scale processes contribute to creep cavitation. Many of the geological works cited interpret that creep cavities are the product of VGBS and form to accommodate strain incompatibilities (Behrmann and Mainprice, 1987; Ree, 1994; Herwegh and Jenni, 2001; Dimanov et al., 2007; Závada et al., 2007; Rybacki et al., 2008; Fusseis et al., 2009; Kilian et al., 2011; Menegon et al., 2015; Précigout and
25 Stünitz, 2016). In the VGBS-based model of Fusseis et al. (2009), cavitation at some grain triple junction is balanced by cavity closure on others. This dynamic model of cavity formation is limited to domains deforming by some form of diffusion creep, and does not account for all reports of creep cavities in geology. In other studies, creep cavitation was linked to the production of crystal defects (Wong, 1990; Rogowitz et al., 2016). It is unclear how these mechanisms relate to each other across rock types or if it is possible for multiple cavitation mechanisms to be active simultaneously.

Despite a growing body of observations on creep cavitation in rocks, some important open questions remain, including:

(i) How ubiquitous are creep cavities in deformed rocks and what combination of deformational processes facilitate their formation?

(ii) How do creep cavities effect an evolving rock rheology and how does this ultimately influence rock deformation?



This contribution addresses question (ii) by examining in detail the nature and occurrence of creep cavities in a mid-crustal ultramylonite from the Redbank Shear Zone (Australia), and furthers our understanding of question (i). We use a sophisticated workflow combining electron microscopy, image analysis, electron back-scatter diffraction (EBSD) and synchrotron-based x-ray nanotomography (nCT) to show that creep cavities can form by multiple mechanisms in one sample. We present a high-resolution map of porosity distribution on the mm scale in an ultramylonite and demonstrate how this porosity evolved during mylonitic deformation.

## 2 Geological setting and sample description

The Redbank Shear Zone (RBSZ) is part of a crustal scale thrust duplex that formed during the Alice Springs orogeny in Central Australia (Teyssier, 1985b). Due to its geometry, where higher-grade shear zones piggy-backed on lower-grade shear zones, the RBSZ has experienced no significant retrograde metamorphic overprint during its exhumation, and the syn-kinematic mineral fabrics and parageneses are preserved (Fliervoet et al., 1997). Micro-fabrics in the RBSZ are therefore ideal for the investigation of transient chemo-physical processes that characterise mid-crustal shear zones.

The RBSZ is a network of shear zones that cascades across scales, with shear zone thicknesses that range from $10^{-3}$ to $10^1$ $m$, displaying a characteristic protomylonite - mylonite - ultramylonite succession (Teyssier, 1985a). This contribution focusses on a quartzo-feldspathic ultramylonite sampled from the amphibolite facies shear zones in the Black Hill area of the RBSZ (Sample BH02, 23°32'46.81"S, 133°25'14.42"E, 350-550 °C; lithostatic pressure, 500 MPa, (Fliervoet et al., 1997)). The sample is a banded ultramylonite that displays a striking and extensive grain-boundary porosity that is hosted in fine grained ($\sim < 20~\mu m$), monomineralic, quartz bands (fig. 1). In addition, the sample shows thick domains of well-mixed polyphase material as well as a network of fine-grained ($\sim 1-2~\mu m$) polyphase layers that envelop large, fractured augen porphyroclasts ($\sim 1~mm$). In general, the sample?s foliation is defined by the monomineralic quartz bands and the thicker polyphase domains, whereby the quartz bands display no signs of boudinage. This work focusses on the microstructure of the quartz bands and the nature of the porosity it hosts. We interpret the disaggregating quartz domains at different stages of dispersal (cf. Kilian et al., 2011) to offer an insight into the locally evolving quartz micro-fabric and an associated porosity.

## 3 Methods

### 3.1 Sample Preparation

We analysed a small sample block, which was cut parallel to the stretching lineation and perpendicular to the foliation (long and short axis dimensions of sample: 22.9/19.4 $mm$) and then polished and carbon-coated for electron microscopy and EBSD. To split the sample along the mylonitic foliation (after electron microscopy), it was pre-cut parallel to the stretching lineation and cleaved in a vice. The split surface was gold-coated.



## 3.2 Data Acquisition and Processing

## 3.3 Microstructural Analysis

A large (41448 x 40282 pixel, at a scale of 1:35.5 ($px : nm$)) back-scatter electron (BSE) map was acquired on a FEI Quanta
FEG 650 SEM operated at an accelerating voltage of 20 kV. This map was stitched from individual images using the Maps
software by FEI.

### 3.3.1 Image analysis

The BSE map formed the basis of a detailed analysis of porosity. In BSE images, pores appear black. They were segmented
using binary thresholding and labelled in Fiji (Schindelin et al., 2012) after a pre-processing workflow was applied to reduce
noise (see supplementary material). Data were visualised with Matplotlib Python libraries (Hunter, 2007). The *Kernel density*
for point features in ESRI's ArcGIS v10.1 software was used in pore cluster analysis. The kernel smoothing factor was auto-
matically calculated with reference to the population size and extent of analysis and contoured based on a $1/4\sigma$ kernel.

From the segmented data, the following parameters were evaluated:

– **Pore size**

Defined as the cross-sectional area (in $\mu m^2$) of a pore.

– **Pore orientation**

Pore orientations were determined by using the long axis of the best fit ellipse and calculating its deviation from the base
of the image. For ease of viewing, the orientation measures were folded along their symmetry axis, $90°$, and presented as
the value $\beta$. For example, $\beta = 0°$ describes a long axis aligned parallel to the base of the image analysed (and orthogonal
to the mylonitic foliation in fig. 1) and $\beta = 90°$ would be orthogonal (and parallel to the mylonitic foliation).

– **Pore shape descriptions**

$$Circularity = \frac{4\pi \left(Pore\,area\right)}{\left(Pore\,perimeter\right)^2} \qquad (1)$$

**Circularity** is a shape descriptor that quantifies the complexity of a shape by linking the area and perimeter. It is
important to note that circularity values are not unique but simply describe a deviation in shape from the area and



perimeter relationship of a circle. A circularity value of 1 describes such a circle and decreasing values can represent either an increase in shape complexity (e.g. a star) or shape elongation, or a combination of both.

$$Roundness = \frac{4\,(Pore\,area)}{\pi\,(Major\,axis)^2} \qquad (2)$$

**Roundness** is a shape descriptor that only quantifies elongation. Used together, circularity and roundness can characterise
pore shape complexity and elongation more correctly.

### 3.4   Electron backscatter diffraction, EBSD

EBSD data were collected on a Zeiss Evo 50 SEM equipped with a Digiview II camera. The sample was tilted to 70°, and a 20 kV accelerating voltage applied [beam current was ≈2.5 $nA$, working distance  11.5 $mm$]. Crystallographic orientation data were obtained with a 0.65 $\mu m$ step size from automatically indexed EBSD patterns in TSL OIM. EBSD data were processed
and plotted with MTEX (Mainprice et al., 2011). Raw data points with <0.1 confidence index (CI) and calculated grains with <10 indexed points were excluded from analysis.

### 3.5   Split sample

Broken surfaces, explored in the SEM, provide insight into pore morphologies and evidence for redistribution of material (cf.
Mancktelow et al., 1998; Fusseis et al., 2009). Images from the split sample were acquired on a Carl Zeiss SIGMA HD VP FEG SEM using a 20 kV accelerating voltage and a ∼6.5 $mm$ working distance. Oxford AZtecEnergy energy dispersive X-ray (EDX) analysis was conducted at an aperture setting of 60 $\mu m$ and used to qualitatively evaluate phase compositions on the split surface. This form of mineral identification was necessary to interpret individual pores in their microstructural context.

### 3.6   X-ray nanotomography

We used an Xradia synchrotron-based nanotomograph at the Advanced Photon Source (beam line APS/32-ID) to acquire 3-dimensional nanotomographic datasets of porous ultramylonitic quartz ribbon bands and assess the potential interconnectivity of pores. A cylindrical sample (20 $\mu m$ diameter x 100 $\mu m$ length) was extracted using a focussed ion beam microscope from a polished thin section wafer that was cut from the sample. In the tomograph, radiographic projections were acquired over a rotation of 180° at 8 keV beam energy and reconstructed to yield 3-dimensional a nanotomographic dataset with a spatial
resolution in the order of 70 nm. Porosity was segmented with binary thresholding and labelled in Avizo Fire. Due to the small sample size and the noisiness of the data, no quantitative analysis was attempted. Volumetric data were visualised in Avizo.





## 4   Results

### 4.1   Micro-fabric domains

The investigated high strain micro-fabric is composed of four microstructural components (fig. 1):

(1) Monomineralic quartz domains are elongated parallel to the foliation, exhibiting a varying degree of coherency as do-
mains. Here coherency is used in the context of the quartz grains spatial distribution, to describe qualitatively the degree to
which quartz grains are aggregated into monomineralic bands vs. dispersed into segregated grains. The most coherent quartz
domains wrap around porphyroclasts and in some cases mantled pophyroclasts. The fringes of the thickest quartz domains
show evidence for the removal of individual quartz grains and their progressive assimilation into neighbouring, poorly mixed
polyphase domains. We assume that thinner quartz domains have advanced further on the path of disaggregating and that the
progression from thicker to thinner quartz domains reflects a progressive microstructural evolution. This assumption does not
presuppose that all thin quartz domains where once very thick, but we consider it highly unlikely that any thin quartz ribbon
was initially just one or two grains (of a 10 - 20 $\mu m$ diameter) wide.

(2) Porosity that can be considered tri-modal: pores on quartz grain boundaries, pores hosted within porphyroclasts, and
cracks that cross cut and run parallel to the foliation. Feldspar porphyroclasts host an intra-crystalline, angular porosity, which
is distinctly different from the inter-crystalline porosity observed in association with quartz (see detailed discussion below).

(3) Well-mixed and poorly-mixed polyphase domains: The finest-grained ($< 1\text{-}2\ \mu m$) parts of the ultramylonite make up
the well-mixed polyphase domains (see left-hand side of fig. 1a). There also exist less well-mixed polyphase domains which
contain disaggregated quartz ($< 10\ \mu m$) (fig. 1b and c). The polyphase domains comprises plagioclase, K-feldspar, mica,
epidote, ilmenite and quartz (fig. 1c).

(4) Porphyroclasts, which are generally either calcic plagioclase or K-feldspar. The K-feldspar porphyroclasts occasionally
display flame perthites. Calcic plagioclase porphyroclasts exhibit what appears to be reaction to fine-grained mantles of K-
feldspar and mica (fig. 1). No quartz porphyroclasts are observed.

### 4.2   Image analysis in the XZ plane of finite strain

Pores were analysed in a representative area (2.1 $mm^2$) of the sample. Figure. 1b shows the full extent of the area used for
both spatial and pore shape analysis. Areas for subset analysis are also marked in figure 1b. Pores in all domains were extracted
and considered in bulk for density analysis. Subsequently, masks were applied to quantify total porosity and analysing pore
shapes in quartz domains separate from porphyroclasts and the polyphase domains. Pores in porhyproclasts and the fine-grained
polyphase domains were analysed together.





### 4.2.1 Spatial distributions of pores

Kernel density analysis demonstrates that the porosity is anisotropically distributed, with a bimodal clustering in respect to domains (figs. 2c and d). The majority of observable pores (86%) exist in direct spatial association with quartz ribbon bands. In quartz domains, the highest density is recorded in the thickest, most coherent ribbons. In the porphyroclast and polyphase domains, the larger feldspar porphyroclasts that have seen the least fracture or reaction to smaller components show the highest density of pores. The total porosity measured in the area shown in figures 2e and f is presented in table 1.

### 4.2.2 Pores in monomineralic quartz

Segmented pores were analysed to identify any systematic changes in pore size, shape and orientation. Figures 3 and 4 show the analysis for the area shown in figure 1b, while figure 5 shows the subset analyses.

### 4.2.3 Pore sizes

It can be seen that pores cover a limited range of values in cross-sectional area (focused strongly around a median value of $0.18\ \mu m^2$) but vary greatly in long axis orientation (fig. 3a). The lower limit of pore area may be controlled by the resolution of the imaging technique. At first inspection there are two maxima in figure 3a: Pores with a low $\beta$ and pores with a high $\beta$. The maximum for low $\beta$ values appears to be more significant.

### 4.2.4 Pore shapes

As stated above, a pore's shape complexity and elongation can be characterised by combining circularity and roundness. When circularity is plotted against roundness, three salient clusters are observed (fig. 3b):

1. Pores with a circular character ($circularity \approx 1, roundness \approx 1$)

2. Pores with an elliptic character ($circularity \approx 1, roundness \approx 0.8$)

3. Pores with a complex shape but only moderate elongation ($circularity \approx 0.3, roundness \approx 0.8$)

Figure 3c shows that when area is plotted against perimeter and coloured for circularity, there are some systematics that can be described by two power law relationships:

$$Area = 0.062 * Perimeter^{1.498} \tag{3}$$

$$Area = 0.072 * Perimeter^{1.081} \tag{4}$$





We assign pores described by these equations to two distinct populations. From figure 3, it is evident that the pores characterised by equation 3 have very high circularity. Furthermore, figures 3b and c suggest that the very circular pores are the smallest (in cross-sectional area). These small, circular pores are then linked by equation 3 to the elliptical pores (elliptical pores having a $circularity \approx 0.8, roundness \approx 0.8$). This relation can be most clearly seen in fig. 3c, where equation 3 de-
scribes pores of a circularity ranging from 1 to $\approx 0.75$. Similarly, equation 4 suggests that all pores with circularity values below $\sim 0.8$ are systematically linked and scale in shape with a power law relationship.

### 4.2.5  Changes in pore orientations

We assume that the long axis of a pore's best fit ellipse is roughly parallel to the orientation of the pore's boundary with the host minerals, and therefore representative of pore orientation. For pores with a circularity less than 1, the feret diameter is
seen to have the same orientation as the long axis of the best fit ellipse (see supplementary fig. 1). Figure 3a shows a variation in pore orientations but does not readily highlight any systematics. However, if figure 3a is considered with figure 3c, it can be seen that pores whose shape is governed by equation 3 generally have a lower value of $\beta$ (see pores with areas $\sim \leq 0.1 \ \mu m^2$). Figure 4 decomposes this observation to show clearly that orientations of the more circular pores ($> 0.8$) rarely exceed $45°$, and predominately assume a low angle to the Z direction of finite strain (fig. 4a). The change in pore orientation at a circularity
of $0.8$ corresponds to the change in the equation governing pore shape (see fig. 3c). There is also a clear propensity for the largest, least circular pores ($> 2 \ \mu m^2$) to be oriented more parallel to the shear plane (fig. 4b).

### 4.2.6  Porosity with a changing quartz microstructure

Spatial analysis of pore occurrences has already shown that pore density decreases with quartz domain thickness (fig. 2d). In combination with microstructural evidence for the disaggregation of quartz domains it is possible to consider the evolution
of porosity congruent with that of quartz domains (fig. 5). It can be observed that the pore shape descriptors change with the quartz microstructure. Firstly, in the thicker quartz domain, both pore populations (described in eqs. 3 and 4) are observed (fig. 5b). In this domain, pores generally have their long axes aligned with the Z-direction of finite strain (fig. 5a). However, both the pore population and orientation change as quartz domains become thinner. It can be seen that in the thinner quartz domain there is an absence of pores from the trend described by eq. 3 (fig. 5d), and that pore orientations become far more variable
(fig. 5c).

### 4.3  Observations of pores in the XY plane of finite strain

### 4.3.1  Pore shapes and orientations

On the broken surface, the porosity present in the thickest quartz domains shows a clear preference to occur along grain-boundaries roughly parallel to the YZ plane (fig. 6a). When the pore morphology is considered with respect to the grain-
boundary arrangement at the pore location, two end-member shapes can be identified. *Firstly* there exist roughly elliptical pores in the YZ plane. These pores can either show an asymmetry, truncating on a flat grain-boundary (e.g. upper most white



arrow in fig. 6a), or be symmetrical about the grain boundary (e.g. upper most white arrow in fig. 6d). *Secondly*, there is an occurrence of angular pores at quartz grain triple junctions (see all yellow arrows in fig. 6).

### 4.3.2 Precipitates on grain-boundaries

The broken surface also reveals information about material redistribution in spatial association with the porosity in the quartz
directly surrounding a plagioclase porphyroclast. The porphyroclast itself shows reaction to more K-rich material, which appears to have a flakey morphology (see the area highlighted as Kfs in fig. 6b). In contrast, nearby quartz grain boundaries are covered in a dendritic material (see lower left-hand side of fig. 6c). The EDX conducted on the broken surface showed the chemistry of the dendrites to be Si-rich, with no other obvious chemical signal. Sharply truncated dendrites (see blue arrow in fig. 6c) seem to preferentially occur on dilatant quartz grain boundaries. Figure 6d highlights textural evidence linking crystal-
lite precipitation (empty blue arrow in fig. 6d) and a dendrite on a dilatant grain-boundary (filled blue arrow in fig. 6d). It is noteworthy that very little evidence for dissolution of quartz can be found. Etch pits were observed only on one site (fig. 6d).

Interestingly, many pores appear empty, but some also seem filled with crystallites. With increasing distance from the porphyroclast there is a transition from the dendritic features on the dilatant grain-boundaries to clusters of crystallites in pores and along grain boundaries (see all blue arrows in fig.6e). At the furthest distances from the porphyroclast, in the quartz domain,
only small amounts of very isolated crystallites are observed (fig. 6b).

### 4.4 Observations of pores in the 3D

Due to a range of technical difficulties, nCT yielded only one dataset that provided insights into a porous quartz layer of about 50 $\mu m$ width. Figure 7 shows a visualisation of labelled pores and highlights interconnected creep cavities in three dimensions. The pores in this layer are mostly oblate and seem to occupy a range of orientations, mostly at high angles to the foliation
(parallel to the top and bottom surfaces of the bounding frame). Most importantly, it is clear from figure 7 that pores are indeed interconnected and not constrained to the polished surface investigated in this study. We consider this proof that at most a small minority of pores observed in figure 1 are formed by plucking during sample preparation.

### 4.5 EBSD analysis

To better understand any potential link between the porosity and the mechanisms accommodating mylonitic deformation in
quartz domains, EBSD analysis was undertaken. The results show clear evidence for crystal plastic processes with the presence of a crystallographic preferred orientation (CPO), sub-grains and the occurrence of lattice distortions (fig. 8 and supplementary fig. 2). These crystal plastic processes are not uniform across the area analysed. It can be seen that the thicker quartz band hosts more features that are considered diagnostic of dislocation creep (fig. 8, subset 1). These processes then become less well-articulated in the thinner quartz band (fig. 8, subset 2). The abatement of lattice distortions and subgrains coincides
with a reduction in grain size and texture strength. The loss of texture strength in the pole figures is further expressed in the





misorientation angle distributions, with the thinner quartz band showing a near random distribution of misorientations (fig. 8, subset 3).

## 5 Discussion

### 5.1 A model for syn-kinematic creep cavitation by different mechanisms

Our study not only advances our understanding of porosity distribution in mid-crustal ultramylonites (see question (i) in the introduction) but also provides critical insight into the mechanisms behind syn-kinematic cavity formation and the associated effects on rock rheology (question (ii) above).

To develop this in detail, we interpret different quartz microstructures in our sample as representing different stages in a progressive evolution ("space for time", see also Fusseis et al. (2006) and Kilian et al. (2011)). If the well-mixed polymineralic
domains are the most mature parts of the studied ultramylonite, monomineralic quartz domains must be considered as relics of an original mylonitic fabric that has been captured in the process of disaggregation. The mechanisms of disaggregation can be observed at the jagged edges of monomineralic quartz domains as well as by comparison of thinner, less coherent, quartz domains with thicker, more coherent, ones. Our EBSD data from these domains suggest that during progressive disaggregation, quartz micro-fabrics with a clear CPO get randomised (fig. 8). This is typically interpreted as indicating a transition from GSI
creep to GSS creep accommodated by viscous grain boundary sliding (Mitra, 1978; Etheridge and Wilkie, 1979; Kerrich et al., 1980; Behrmann and Mainprice, 1987; Závada et al., 2007; Kilian et al., 2011; Herwegh et al., 2014; Menegon et al., 2015; Viegas et al., 2016).

Our data show how a syn-kinematic porosity can be associated with this inferred change in rheology, and the formation of a quartzofeldspathic ultramylonitic micro-fabric. In our sample, quartz domains are associated with an overt porosity and we con-
sider the pores in the quartz domains to be creep cavities (Riedel, 1987). We claim that the cavities evolved syn-kinematically with both the microstructure and the dominant deformation mechanisms, finding that the disintegration of quartz domains is a result of the emergence of creep cavities. These creep cavites have two distinct populations and formed hierarchically. We integrate our findings in a model (fig. 9) where Zener-Stroh cracking produced small cavities in domains deforming by GSI creep (fig. 9b, Time 1). We infer that their formation promoted fluid ingress, which in turn lowered the adhesion and cohesion
of grain boundaries (cf. Billia et al., 2013). The addition of a fluid locally increased the contribution of VGBS to strain energy dissipation, which led to the formation of a second population of creep cavities (fig. 9b, Time 2) (Fusseis et al., 2009).

Image analysis constrained the two populations of creep cavities in detail (fig. 3c):

(i) Cavities with a perimeter smaller than about $\sim 2\mu m$ with long axes that have a low angle to the YZ plane of finite strain (figs. 3c and 4a) and are governed by the power law relation in equation 3.

(ii) A population of larger cavities that has initially jagged or incised perimeters and generally elliptic shapes (figs. 3b and c). The long axes of these larger cavities exhibit a wider variation in orientation (fig. 4a). However, the largest of these



cavities tend to be aligned with grain boundaries that are parallel to the shear plane (fig. 4b). This population is governed by the power law relation in equation 4.

In our sample, thicker, more coherent quartz domains show more of the former population, and thinner, less coherent quartz domains contain only cavities of the latter characteristics (fig. 5). It can also be observed that congruent with the change in cavity characteristics and domain width, there is a randomisation of the CPO present (fig. 8c). We interpret these observations to indicate that larger cavities are associated with VGBS.

Syn-kinematic creep cavitation by VGBS is not a new idea in geology: several microstructural (Behrmann and Mainprice, 1987; Herwegh and Jenni, 2001; Závada et al., 2007; Fusseis et al., 2009; Kilian et al., 2011; Menegon et al., 2015) and experimental works (Ree, 1994; Dimanov et al., 2007; Rybacki et al., 2008; Précigout and Stünitz, 2016) invoke the process. Experimental work by both Dimanov et al. (2007) and Rybacki et al. (2008) show that pores generated during deformation by VGBS have a complex, angular and elongated shape. This agrees well with our observations from increasingly disaggregated, thinner quartz domains (fig. 5, subset 2) with their weakened CPO (fig. 8, subset 2). As this type of cavity formation has been well discussed in previous contributions we will not examine this further.

In some contrast, geological descriptions of creep cavities that formed by Zener-Stroh cracking are still rare (Rogowitz et al., 2016). In quartz domains of our sample that deformed dominantly by GSI creep, the population of very small cavities do not occur at grain triple junctions but rather seem to nucleate along grain-boundaries that are aligned in the YZ plane of finite strain (figs. 4a, 5 (subset 1) and 6, all white arrows). Boundaries in these orientations are mechanically unlikely to experience significant sliding, which suggests that VGBS is not conditional for cavity formation and alternative mechanisms might be relevant. In material science, it has been shown that creep cavities in an environment that is characterised by work hardening can nucleate by the coalescence of dislocations. This will occur where crystallographic slip bands intersect with grain-boundaries or grain-boundary precipitates allowing dislocations to pile up, thus forming Zener-Stroh cracks (Stroh, 1957; Bauer and Wilsdorf, 1973). On the basis of our observations we speculate that cavitation by Zener-Stroh cracking could have provided an initial porosity in monomineralic quartz domains that emerged directly out of GSI creep and potentially played an important role in the rheological evolution of the ultramylonite.

## 5.2 Nucleation of creep cavities during GSI creep

Despite evidence provided by cavity shapes and orientations (fig. 3,4,5 and 6) unequivocal proof of Zener-Stroh cracking is difficult. For the Zener-Stroh mechanism to act, the material must have an abundance of crystal defects produced by deformational work (Stroh, 1957). In our sample, the presence of a CPO, sub-grains and the occurrence of lattice distortions in the thick quartz domains are evidence for the activity of dislocation creep processes and hence the production of crystal defects in the same microstructural domains that host creep cavities. Our analysis further shows that the orientation of the dominant slip in these domains is aligned with the X direction of finite strain (fig. 8, subset 1), i.e. favourably for contributing to cavitation by Zener-Stroh cracking. Grain-boundaries at high angles to the X direction (i.e. those in the YZ plane) could provide the obstacles required for dislocations to pile up. Evidence for quartz grain-boundary porosity in association with high dislocation




density has been reported in previous studies from ultramylonites (Shigematsu et al., 2004; Behrmann, 1985; Rogowitz et al., 2016). Recently, Rogowitz et al. (2016) explicitly invoked the Zener-Stroh mechanism to explain small grain-boundary cavities next to dislocation pile ups. In the light of these studies and despite that fact that we can only provide indirect evidence, its seems reasonable to speculate that the process of cavity nucleation could be that of Zener-Stroh cracking.

A dislocation-driven cavitation would require an appropriate density of dislocations to be present for a creep cavity to nucleate. It follows that cavity production consumes defects, and in this way it is a process that may directly compete with other recovery processes such as sub-grain wall formation. Our EBSD observations of cavities and sub-grains suggests that both coexist. (Kilian et al., 2011) has demonstrated the importance of sub-grain wall formation in the disaggregation process of quartz domains in quartzo-feldspathic ultramylonites that have experienced similar geological conditions as those investigated

here. From an irreversible thermodynamic perspective, integrating cavitation as a dissipative component of dislocation creep would expand our current understanding of a crystal's internal entropy production (see eq. 1 of Huang et al. (2009)). This would alter equation 18 of (Huang et al., 2009), which describes the total rate of reduction ($\rho^-$) of the dislocation density at a steady state ($\rho$). Integrating Zener-Stroh crack formation as a $\rho^-$ mechanism would yield:

$$\frac{d\rho^-}{\gamma} = \frac{d\rho^-_{DRX}}{\gamma} + \frac{d\rho^-_{DRV}}{\gamma} + \frac{d\rho^-_{CAV}}{\gamma} \tag{5}$$

Where $\frac{d\rho^-_{DRX}}{\gamma}$, $\frac{d\rho^-_{DRV}}{\gamma}$ and $\frac{d\rho^-_{CAV}}{\gamma}$ are the dislocation annihilation rates due to dynamic recrystallisation, dynamic recovery, and creep cavitation respectively.

Rogowitz et al. (2016) have recently demonstrated in a fine-grained calcite ultramylonite ($\sim 3\ \mu m$) that recovery can occur without the formation of sub-grain walls. Internal strain is recovered by extensive glide and dislocation networks characteristic of cross-slip and network-assisted dislocation movement. In conjunction with this mechanism, Rogowitz et al. (2016) also

observed Zener-Stroh crack formation. This may outline a scenario where cavity production dominates over sub-grain wall formation.

Where we infer Zener-Stroh cracking to be responsible for cavity nucleation, Hippertt (1994) proposed an alternative model to explain a concentration of pores on grain-boundaries at high angles to the fabric attractor in a sheared micaceous quartzite (see fig. 10 in Hippertt (1994)). Hippertt (1994) suggested that the initial porosity is loosely connected to preferential dissolution

of quartz at sites where dislocation tangles intersect grain boundaries (cf. Wintsch and Dunning, 1985). In both the sample of this study, and of Hippertt (1994), there is a clear systematic link between the pore orientation and the bulk finite strain. Considering the inferred orientations of compression and extension in our sample, preferential dissolution should be expected orthogonal to the observed cavities (see etch pit formation in fig. 6d), i.e. creep cavities in our sample open at sites that would be favourable for precipitation, not dissolution. We therefore consider the model of Hippertt (1994) as incompatible with our

observations.



## 5.3 The role of creep cavities in the activation of GSS creep

The rheological evolution of mid-crustal ultramylonites to GSS creep is prompted by the establishment of a very fine grain size. In ultramylonites that deform by GSS creep, grain growth is usually inhibited by the presence of secondary phases, a process called Zener pinning (Herwegh et al., 2011). Where syn-kinematic creep cavities control fluid transport in ultramylonites (Fusseis et al., 2009; Menegon et al., 2015), cavities should also control secondary phase precipitation and hence directly influence Zener pinning (Herwegh and Jenni, 2001). This invites a discussion on how creep cavitation could influence Zener pinning and thereby facilitate the transition of a rock's rheology from GSI to GSS creep. A critical step in the rheological evolution of the ultramylonite investigated here is the transition from GSI to GSS creep in disintegrating quartz domains.

Conspicuously, in the monomineralic quartz domains secondary minerals are generally absent. We speculate that fluid-filled creep cavities will have affected grain boundary migration directly by acting as pinning phases, therefore arresting grain sizes at sub-equilibrium dimensions and promoting the transition to VGBS. This idea can be further explored by combining our data with the Zener parameter, which quantifies the influence of second phases on rheology (Herwegh et al., 2011):

$$Z = \frac{d_{sp}}{f_{sp}} \tag{6}$$

where $Z$ is the Zener parameter, $d_{sp}$ is the size and $f_{sp}$ is the volume fraction of the secondary phases. Equations 3 and 4 give an empirical indication into the value of $d_{sp}$ for creep cavitation via Zener-Stroh cracking and VGBS, respectively. The dynamic nature of creep cavitation suggests that $d_{sp}$ is not constant but varies between a maximum and a minimum that can be taken from figure 3a. However it is unclear how $f_{sp}$ would evolve with the different cavity formation mechanisms. Any porosity derived exclusively from dislocation creep would be expected to have a characteristic spacing between pores on a grain-boundary, dictated by the crystal volume and the amount of strain that an individual slip system can accommodate. Therefore in this scenario $f_{sp}$ would be directly linked to this characteristic spacing. On the other hand, values of $f_{sp}$ generated by porosity linked to VGBS would have a different character. The volume fraction in this case may be linked to a space problem, where the amount of dilatancy is limited by the surrounding grains. In either case, cavitation should be considered as mechanism that is capable of evolving the Zener parameter and hence the rheology of a domain from GSI to GSS.

Our model ties in with more recent experimental observations by Précigout and Stünitz (2016), who also idenify creep cavitation as a means of producing domains that deform by GSS creep. In contrast to our results Précigout and Stünitz (2016) discuss the deformation of clinopyroxene embedded in an olivine matrix, where phase mixing occurs in clinopyroxene tails. This process is interpreted to be initiated by micro-cracking. Précigout and Stünitz (2016) advocate a model where the nucleation rate of secondary phases is high. New phases are precipitated simultaneously with micro-cracking and each new cavitation site becomes filled with new phases, which suppresses the development of a CPO. An implication of this model would be that the rate of precipitation is fast enough to inhibit the development of a dynamic granular fluid pump (Fusseis et al., 2009). On the other end of the spectrum, our observations highlight a scenario where the rates of precipitation are so slow that fluid-filled cavities remain unfilled. Evidence of quartz precipitation is possibly observed in the form of Si-rich grain boundary features (fig. 6, see all blue arrows), but in our interpretation any precipitation is volumetrically not significant enough to fill cavities.





Another major difference between the two models is that our model does not require brittle fractures to initiate the disaggregation of a monomineralic domain. Our observations also do not necessitate the establishment of a dynamic granular fluid pump in the monomineralic quartz domains but it seems likely that if a dynamic granular fluid pump was active in the polyphase domains, the quartz domains would participate in some way. The results of our work probably showcase an example where the

5 nucleation of phases is not kinetically or energetically favourable.

### 5.4 A lack of boudinage but maintenance of strain compatibility

A striking feature of the ultramylonite is the lack of any evidence for boudinage in quartz layers. This implies that either the syn-kinematic viscosity contrast between the polyphase and the quartz domains was small (Smith, 1975; Viegas et al., 2016), or that the quartz layer was not able to achieve localisation because the local temperature fluxes were efficiently dissipated (Peters

et al., 2015). Boudinage by either of these processes is considered a ductile instability, where irrecoverable change occurs and grows over time (cf. Peters et al., 2015). In our sample a syn-kinematic porosity is observed and can be considered itself a ductile instability. As discussed above, creep cavitation by Zener-Stroh would be a dissipative feature of dislocation creep that would act to lower the internal energy of a grain. In thermodynamic terms creep cavities could act as an energy sink. From a micro-mechanical perspective, a syn-kinematic porosity offers the possibly of lowering grain-boundary adhesion and cohesion

as fluid is drawn to low stress sites (Fusseis et al., 2009; Billia et al., 2013), thereby compromising the rheological integrity of the monomineralic quartz domains and promoting sliding. Therefore, it may be the case that cavity formation in quartz domains inhibits strain localisation via boundinage by increasing the contribution of VGBS, which in turn accommodates the extension of quartz layers, facilitating the quartz bands' ultimate demise.

### 6 Conclusions and outlook

In this study we utilise a workflow of SEM based techniques and synchrotron x-ray nanotomography to rigorously examine the nature and occurrence of a grain-boundary porosity found in recrystallised quartz ribbons of a quartzofeldspathic ultramylonite. We find that the porosity developed syn-kinematically from the deformation mechanisms active in quartz and the pores can thus be considered as creep cavities. We propose a model of hierarchical creep cavity formation that has implications for both the mircostructural and rheological maturation of an ultramylonitc fabric. We interpret based on the orientation of creep cavities

and the crystallographic texture of quartz domains that Zener-Stroh cracking is responsible for the initial nucleation of creep cavities. The opening of creep cavities promotes the ingress of fluids to sites of low stress, and the local addition of a fluid lowers the adhesion and cohesion of grain-boundaries promoting VGBS. The increased activity of VGBS is documented in the thinning of quartz domains. In thinner quartz domains both the texture weakens and cavities become more complex, eventually elongating. We suggest that cavitation at this stage of the quartz microstructural evolution is governed by VGBS. Zener-Stroh

cracking can be directly linked to crystal plasticity, and our observations therefore potentially point to a wider significance of creep cavitation in mylonitic deformation. It remains unclear if the emergence of Zener-Stroh cracking is contingent on quartz becoming the locally stronger phase. This would restrict the model presented here to scenarios where some fine grained



mixtures have already emerged. Most importantly our findings document a micro-mechanical path for clustered quartz grains to be dispersed into an well-mixed phase mixture.

Both of the invoked creep cavity formation mechanisms are well know from material sciences and are intimately linked to ductile failure in metals and ceramics (Bauer and Wilsdorf, 1973; Gandhi and Ashby, 1979; Riedel, 1987; Shigematsu et al.,

5    2004). Our model points to the coeval activity of both mechanisms in mid-crustal ultramylonites. This raises questions about how these creep cavities interact? While it is unclear if natural samples can reveal such transient aspects, it is clear that such questions of critical importance in furthering our understanding of mylonitic processes.

*Data availability.*  High resolution BSE image available from J. Gilgannon (james.gilgannon@geo.unibe.ch)



**Table 1.** Porosity data from fig. 2

| Domain | Number of pores in domain | % of total porosity (%) | Absolute porosity ($\mu m^2$) | Porosity presented as % of total area of fig. 1b (%) |
|---|---|---|---|---|
| Quartz | 6991 | 86 | 1515 | 0.07 |
| Porphyroclast + Polyphase | 1138 | 14 | 247 | 0.01 |



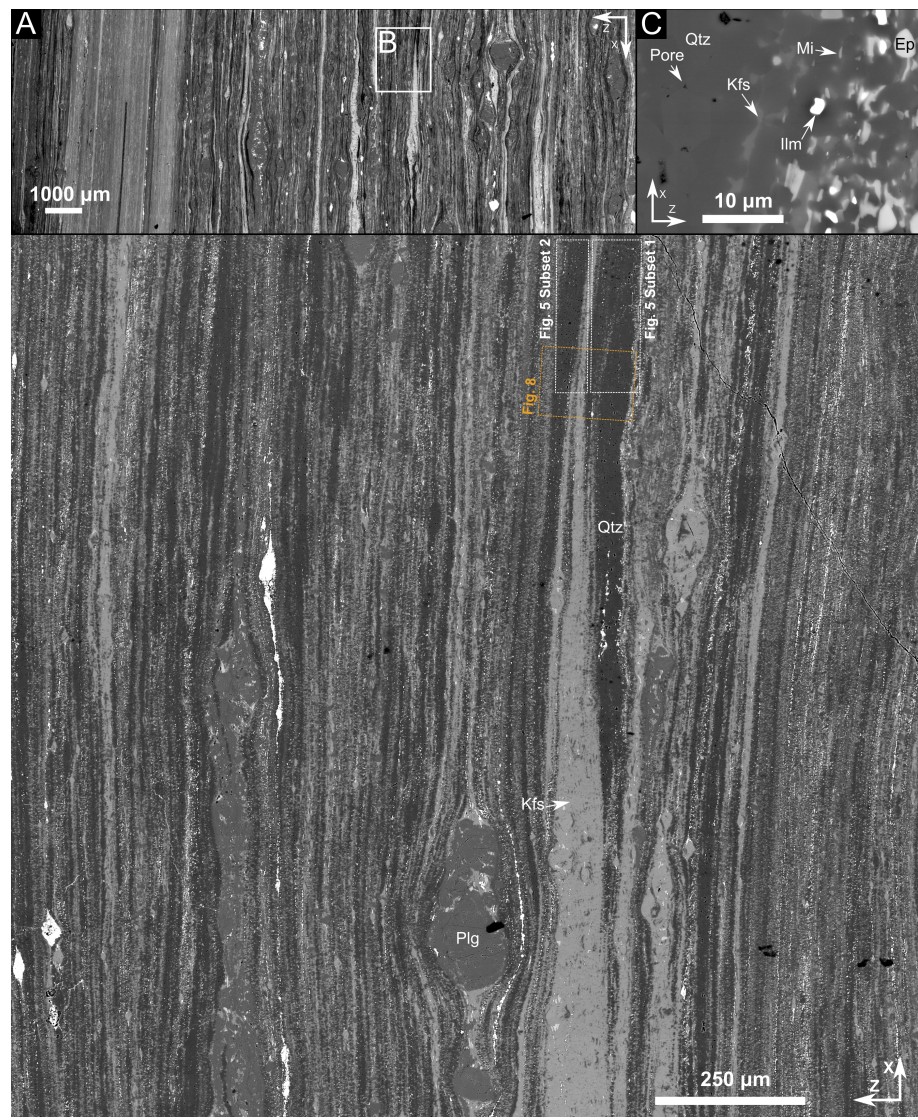

**Figure 1.** BSE images from the quartzofeldspathic ultramylonite. Fig. 1a shows the overall strain gradient in the sample, with the highest strain domain found on the left of the image. Fig. 1b is a high-resolution BSE SEM mosaic of a representative area of the sample (41448 x 40282 pixels, scale of 1 px : 35.5 nm). All results presented for pore shape and orientation analysis are from the area of fig. 1b. The greyscale values identify minerals as follows: black = Porosity, dark grey = Qtz, grey= Plg, light grey = Kfs, bright = accessory phases. Fig. 1c presents the edge of a disaggregating quartz domain and highlights the minerals present in the poorly mixed polyphase domains.





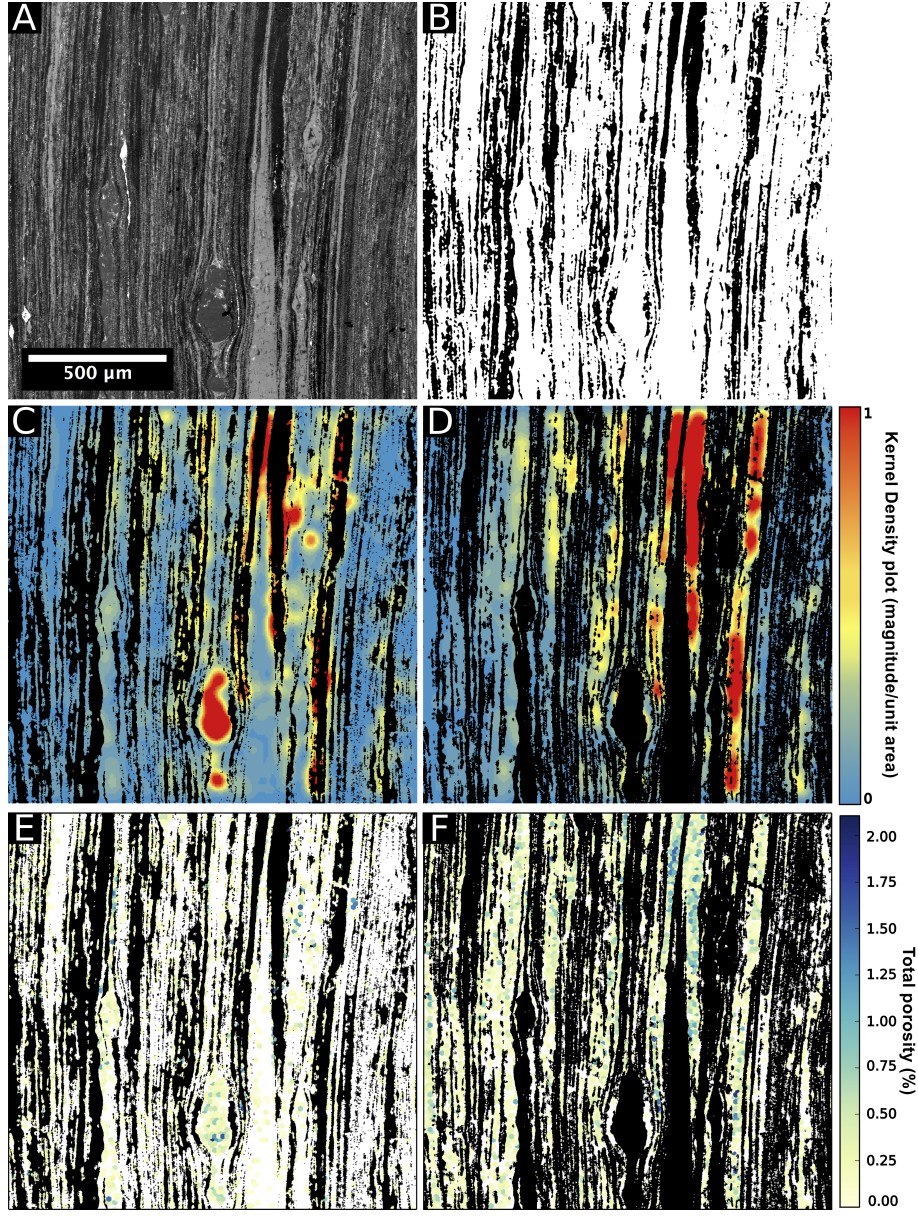

**Figure 2.** The spatial distribution of porosity in figure 1b. In the BSE mosaic of fig. 1b, 8129 individual pores were identified. For reference, fig. 2a displays the micro-fabric of the area analysed. Fig. 2b, is the mask used for distinguishing microstructural domains for analysis: quartz is coloured black. Fig. 2c and d, present masked kernel density analysis, highlighting regions of pore clustering. Fig. 2c is masked to remove all quartz and fig. 2d is masked to only show the quartz. The clustering of pores is most prevalent in the thickest, most coherent, quartz domains and in the largest feldspar porphyroclasts. Fig. 2e and d are hexbin plots that are coloured to show the absolute porosity per hexbin (masked in the same fashion as fig. 2c and d)(see also Table 1 for values per domain).




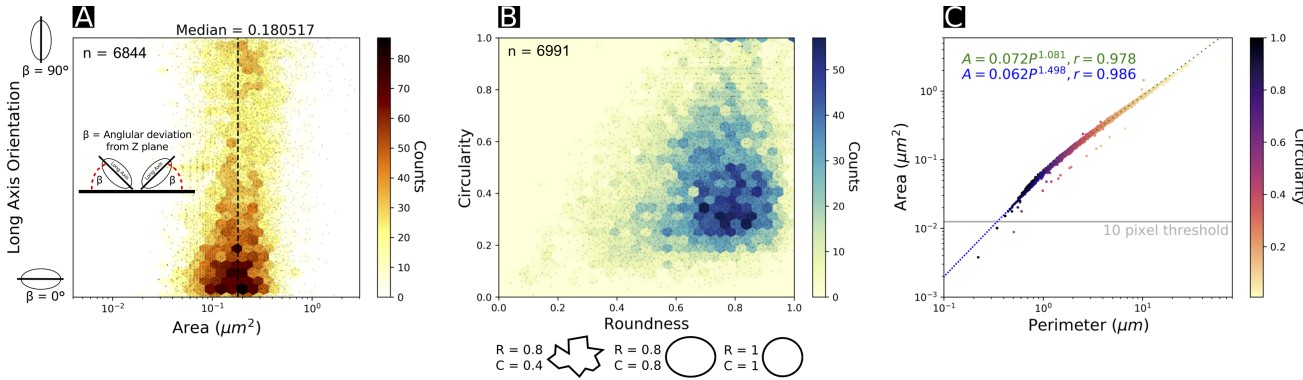

**Figure 3.** Plots of relationships in pore shape, size and orientation for all pores observed in association with the quartz domains of fig. 1b. Due to the large sample size, pertinent clustering in data is more clearly observed when presented in hexbin plots (actual data points are overlain). In figs. 3a and b, the hexbin plot colouring displays the number of data points contained within each hexbin. Fig. 3a presents area as a function of pore long axis orientation ($\beta = 0°$ is parallel to the Z plane of finite strain). Circular pores (circ = 1) are excluded from fig. 3a because the long of a circle will not have a unique or meaningful orientation. Fig. 3b compares each pore?s circularity with its roundness. One large cluster (circ = 0.4, round =0.8) and two minor clusters (circ = 1, round =1; circ = 1, round =0.8) are observed. Fig. 3c shows the relationship of pore perimeter with increasing cross-sectional area. Each data point is additionally coloured for its circularity. Two power law trends are identified in fig. 3c. See text for further discussion.





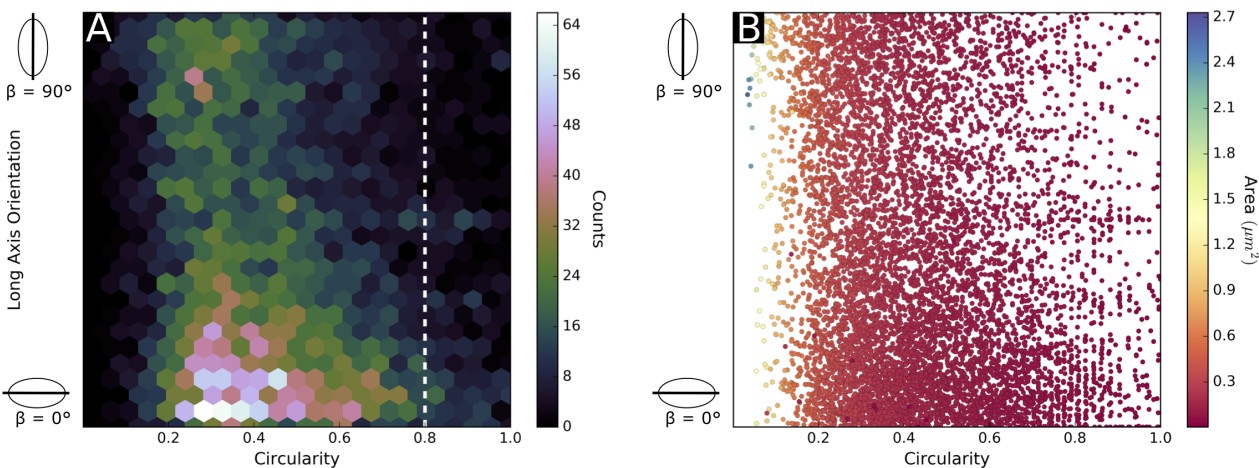

**Figure 4.** The link between pore orientations, pore sizes and their shapes in detail. Fig. 4 only presents data for pores with circularity values <1. A delineation (circ = 0.8) is presented to show the change in power law relations observed in fig. 3c. Fig. 4a shows that with decreasing circularity there is an increase in the variability of $\beta$ values. Fig 4b presents the same data as fig. 4a but with the data points coloured for cross-sectional area. The largest pores are observed to mostly have high $\beta$ values.



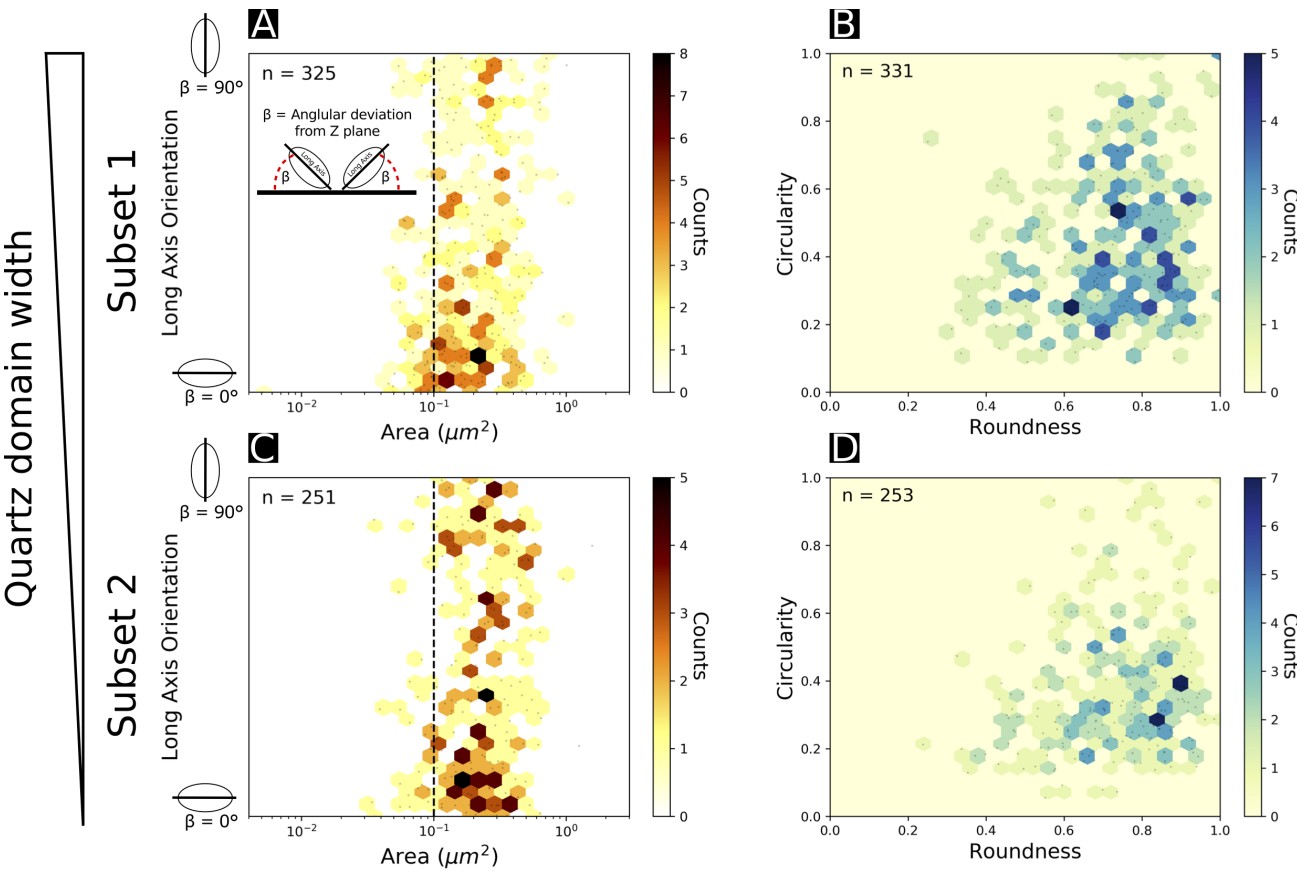

**Figure 5.** Pore analysis for subsets 1 and 2 shown in fig. 1b, corresponding to decreasing quartz domain width and increasing quartz dispersion. Fig 5a and c show that with decreasing quartz band thickness, there is an increase in the range of pore orientations observed. Fig. 5b and d demonstrate that this change is also concordant with a change in the pore shapes from roughly elliptical to more complex. More specifically, it can be seen in fig. 5b and d, that with decreasing domain width there is a loss of circular and elliptical pores.





**Figure 6.** Secondary electron images from broken surfaces of quartz domains. White and yellow arrows point, respectively, to pores on grain-boundaries with a low angle to the YZ plane of finite strain and to pores at grain boundary trip junctions. Blue arrows highlight 'precipitation' features, with the empty blue arrow in fig. 6d identifying crystallites interpreted to be incipient precipitates found on a less dilatant grain boundary.





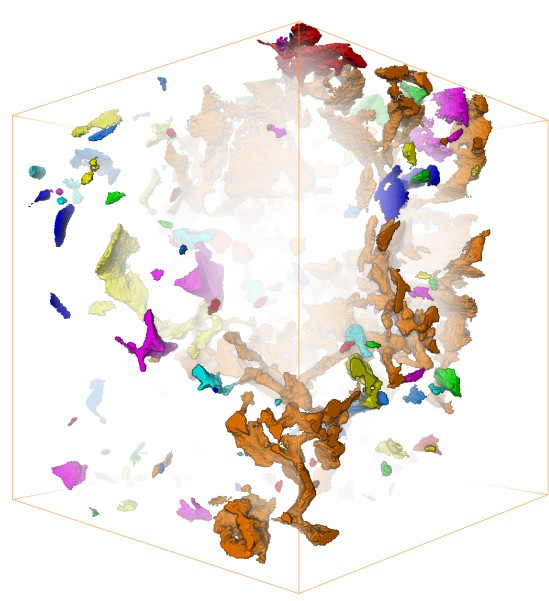

**Figure 7.** 3-dimensional rendering of cavities segmented from a nanotomographic dataset. Individual pores are coloured, note large interconnected pore cluster in orange. Dimension of the cube is 700 $voxel^3$, with a voxel size of ∼35 nm. The top and base of the cube are parallel to the mylonitic foliation. The figure indicates the oblate shape of most cavities and proves that they are indeed 3-dimensional features.



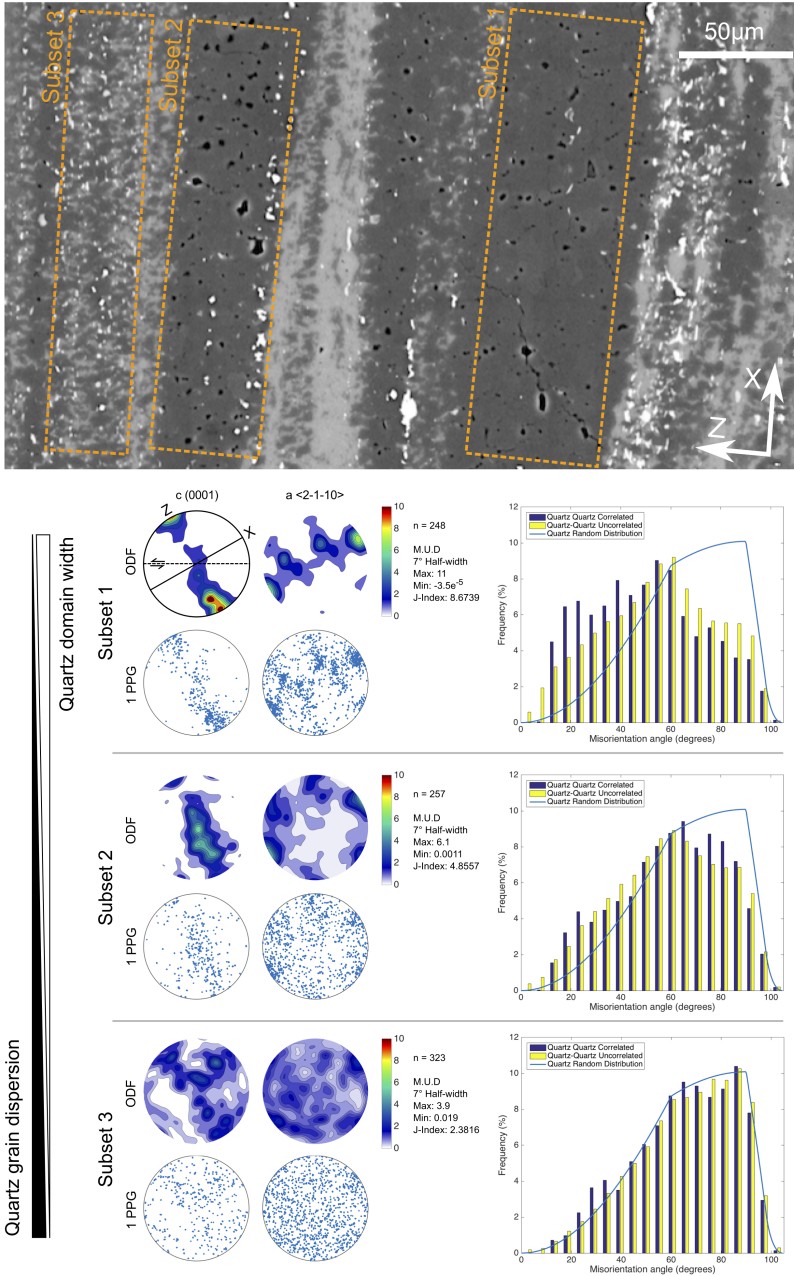

**Figure 8.** EBSD analyses of quartz domains. The EBSD analyses were preformed in the same quartz bands used for subset image analysis (see fig. 1 and 5). EBSD data are presented in equivalent subsets corresponding to decreasing quartz domain width and increasing quartz dispersion. A clear CPO is observed in the pole figure analysis of subset 1 with two c(0001) maxima and two corresponding a<2-1-10> maxima. As quartz domain thickness decreases there is a randomisation of the CPO, highlighted by the decrease in the J-Index and supported by the shift to a near random distribution of the misorientation angle histogram of subset 3.



## A Model for Hierarchical Cavity Formation

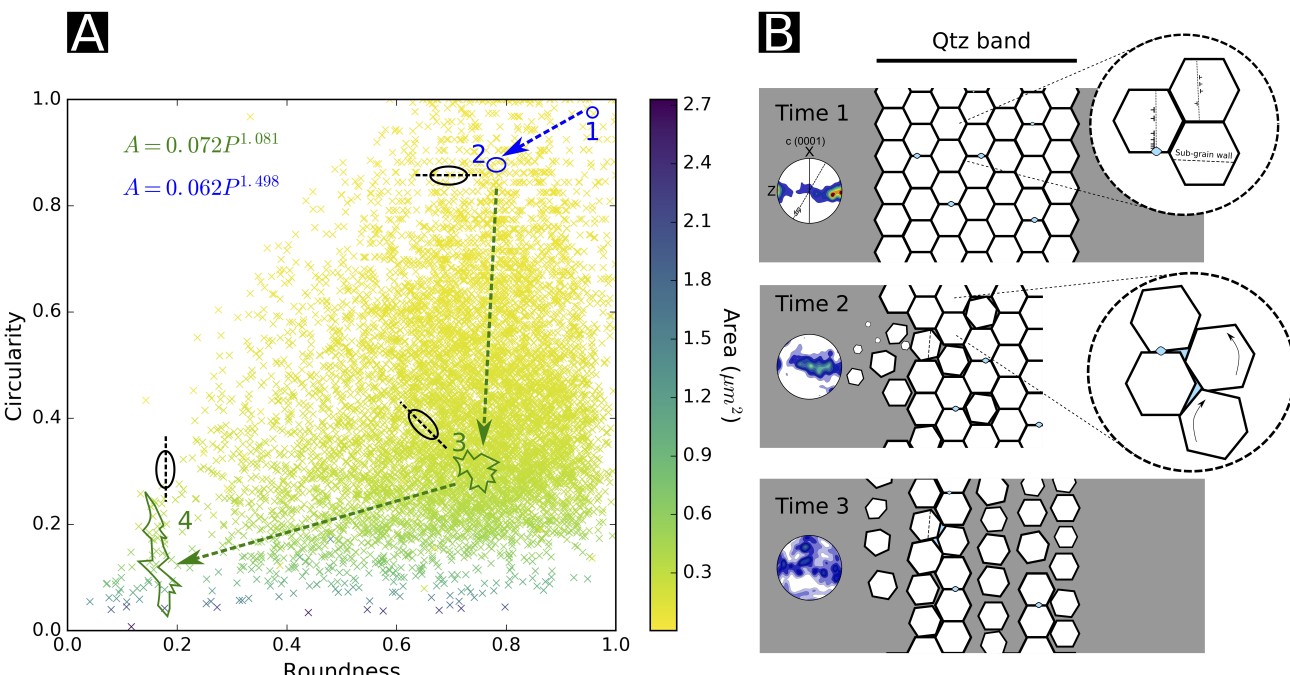

**Figure 9.** A model for synkinematic creep cavity formation by two different mechanism. Fig. 9a graphically represents the inferred trajectory of a creep cavity's life cycle. Initially creep cavities grow from circular (1) to elliptical cavities (2). These cavities generally have long axes aligned in the Z direction (when viewed in the XZ plane). As cavities become larger they develop more complex shapes and rotate (3), ultimately elongating and becoming more aligned with the shear plane (4). Fig. 9b schematically integrates our observations of cavity evolution with the evolution of the microstructure and the associated micro-mechanisms. First cavities form, in quartz domains deforming by GSI creep, by the Zener-Stroh mechanism (fig. 9b, Time 1). As the grain boundary strength is weaken by the ingress of fluids into cavities, VGBS is promoted. This increase in the contribution of VGBS drives the production of new creep cavities of a more complex shape (fig. 9b, Time 2). It is the production of creep cavities that initiates the increased contribution of VGBS and ultimately prompts a switch to a GSS creep. See text for details.

*Competing interests.* No competing interests are present

*Acknowledgements.* We would like to thank Nicola Cayzer (Edinburgh Materials and Micro-Analysis Centre) and Natasha Stephen (Plymouth Electron Microscopy Centre) for their help with the acquisition of SEM data. James Gilgannon would like to thank Marco Herwegh for invaluable discussions about Zener pinning, Cees-Jan De Hoog for council on fluids and phase precipitation, and finally Alfons Berger for EBSD procurement. This work was financially supported by the School of Geosciences, University of Edinburgh. Use of the Advanced Photon Source at Argonne National Laboratory was supported by the U.S. Department of Energy, Office of Science, Office of Basic Energy Sciences, under Contract No. DE-AC02-06CH11357.



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
