# Peer review of "Hierarchical creep cavity formation in an ultramylonite and implications for phase mixing"

_Solid Earth, 2017_

## Referee Comment (RC1) · J. Précigout (Referee) · 25 Sep 2017

Dear editors and authors,

This study provides detailed documentation of the distribution of micro-cavities in a quartzo-feldspathic ultramylonite from the Redbank Shear zone (Australia). Based on interesting observations, the authors highlight different processes of creep cavitation that may account for strain-induced phase mixing in natural rocks, which is a sine qua non condition to highly reduce grain size and promote intense strain localization. I found the paper very well written, up-to-date and focusing on processes that need to be properly understood regarding their fundamental implications. Although some parts of the manuscript need to be completed and reorganized, the results are convincing

and the discussion is coherent, giving rise to a significant advance in this field of research. I therefore strongly recommend this paper to be published, provided that my comments/concerns (in appended .pdf) are addressed by the authors.

With my Best Regards

Jacques Précigout

Please also note the supplement to this comment:
https://www.solid-earth-discuss.net/se-2017-96/se-2017-96-RC1-supplement.pdf

**Supplement:**

Review of « **Hierarchical creep cavity formation in an ultramylonite and implications for phase mixing** » by James Gilgannon, Florian Fusseis, Luca Menegon, Klaus Regenauer-Lieb and Jim Buckman

For publication in Solid Earth
Research article N° se-2017-96
Paper reviewed on Septembre 25th, 2017

Dear editors and authors,

This study provides detailed documentation of the distribution of micro-cavities in a quartzo-feldspathic ultramylonite from the Redbank Shear zone (Australia). Based on interesting observations, the authors highlight different processes of creep cavitation that may account for strain-induced phase mixing in natural rocks, which is a *sine qua non* condition to highly reduce grain size and promote intense strain localization. I found the paper very well written, up-to-date and focusing on processes that need to be properly understood regarding their fundamental implications. Although some parts of the manuscript need to be completed and reorganized, the results are convincing and the discussion is coherent, giving rise to a significant advance in this field of research. I therefore strongly recommend this paper to be published, provided that my comments/concerns below are addressed by the authors.

With my Best Regards

Jacques Précigout

Detailed comments/concerns:

Page 1-abstract: The object and techniques used have to be summarized in the abstract. In its present form, we have no idea of what the authors did.
Page 1-Line 1: Spelling « quartzo-feldspathic »
P1-L14: spelling « conditions »
P3-L20: spelling « sample's foliation »
P3-L27: More information about the thicknesses of carbon coating have to be given here. Actually, in an ideal case, it is not recommended to use carbon coating for EBSD analyzes, but sometimes, it is better to add a few nanometers to avoid « charging » effect. So what thickness did you put on your sample block ?
P5-L10: Please add the reference of Bachmann et al. (2010) for MTEX
P5-L11: Some precisions are here required concerning the minimum number of indexed points per grain. Is that in one row, several rows or for the whole grain? We commonly use a minimum threshold of 5 consecutive pixels in several rows.
P8-L1: I would add a comma after « domains »
P9-L1: The figure 6d is called before figure 6b. Please, make sure that the figures are properly called in the manuscript (in successive order).
P10-L23: The figure 9a is not called anywhere.
P11-L20: Please add the reference of Kassner and Hayes (2003) after « dislocations ».
P12-L13: « Kilian et al. (2011) has… » and not « (Kilian et al., 2011) has… ».
P13-L9: Place « secondary minerals are generally absent » before « in the monomineralic… », and not after.
P13-L24: Spelling « identify »
P13-L25: add a comma after « results ».

P13-L29: It would be interesting to discuss our paper here (Précigout et al., 2017, Nature communications), which deals with the relationships between creep cavitation and phase nucleation in ultramylonites. The authors also have to discuss the recent paper of Cross and Skemer (2017, JGR solid Earth). Moreover, I have a question about your sentence claiming that fluid-filled cavities remain unfilled: if phase nucleation does not arise from creep cavitation, how do you explain the transition from GSI to GSS creep by phase mixing? In our paper in 2016 (Précigout and Stünitz, 2016), we do not claim that phase nucleation necessarily follows cavitation. We just say that phase nucleation is fast enough to maintain grain size small and randomize the olivine fabric. Some cavities may remain unfilled, particularly if the fluid is undersaturated.

P14-L11: add a comma after « sample ».

P15-L4: Spelling « known »

Figure 1: Giving the GPS point is not enough, the authors have to provide a simplified map of Australia that locates the sample area. I would also recommend to add a picture of the outcrop. The figure 1C is not located. The figure 1B is not labelled.

Figure 2 (caption): I think it is « figure 2e and f », not « figure 2e and d ».

Figures 6 and 7: these two figures arrive too late in the manuscript. They should appear after figure 1, particularly to show the microstructural features of pores. The text will have to be changed accordingly. By the way, the figure 1C has to be shown with figure 6. The figure 7 also demonstrates that the authors documents 3D features coeval with rock deformation. It has to be given before going into details concerning the distribution and shape of micro-cavities.

Figure 6 (caption): please details the sub-figures (a, b, c, etc.). I am not sure that the figure 6e is necessary.

Figure 8: The EBSD maps have to be shown in the manuscript (not in supplementary material), at least to show the sub-grains. I would recommend to show the three of them. Furthermore, the c axes have to be spelled between square brackets (« [0001] ») and the <a> axes are commonly indicated using <11-20>. The pole figure texture is based on, but not the ODF. Please change « ODF » by « texture » (or equivalent). Please provide the Mindex, as well. That will definitely confirm your point about the distribution of misorientation angles. Use « uniform » instead of « random » in the figure legend.

---

## Referee Comment (RC2) · L. Morales (Referee) · 6 Oct 2017

In this paper the authors performed a detailed microstructural analysis of a sample of quartz-feldsphatic ultramylonite from a shear zone in Central Australia in order to understand the origin of creep cavities ubiquitously observed in the studied sample. For that the authors have used multi-technique workflow combining techniques of electron microscopy and x-ray nanotomography. Through these different approaches the authors were able to visualize not only where the porosity is concentrate in the studied sample, but also determine the orientation and shape of these pores. I liked the idea of a "two" step cavity formation and the role of creep cavitation on the transition from grain insensitive to grain sensitive rheology in this ultramylonite example. The paper is very well written, easy to follow, the figures are really good and the paper is in a good

shape to be published. Nevertheless the authors may want to consider the comments below before publication:

1) Although I like the idea of Zener-Stroh cracking mechanism to explain the initial porosity in the quartz rich bands, the evidence provided is not totally convincing because of the lack of TEM analyses in the studied sample. The TEM imaging in this case is really necessary because one has to be able to see the dislocations aligned against some sort of "barrier" (a grain boundary, or particular slip plane), where they would piled up and eventually coalesce to form voids and cracks. This is obviously not very easy but would be a more convincing evidence for the activation of this mechanism during the deformation of the quartz bands and the cavities. The authors also have to keep in mind that in quartz one will be never sure if the dislocations pile up to form the porosity of if the porosity (and the stress concentration around it) will be the place where the dislocations are nucleated, because we cannot see dislocations moving. Another possible way to tackle better this problem would evolve, for instance, the detailed EBSD mapping around the pores to see if there any evidence of more distorted lattice around the voids or somewhere in the grains;

2) The authors mentioned that roughly the porosity is generated in grain boundaries aligned with the YZ plane of finite strain. From the graphics of Fig. 3, the predominant porosity shape is rather irregular, and although there is a predominance of porosity long axes parallel to Z (Fig. 3a), I did not understand the relation to Y, considering that the analyses were performed in the XZ section. Maybe there is some piece of information missing about the calculation of the Y-axis (for instance, as a cross-product of the long and short axes extracted from the maps). Or maybe the authors should include analysis in an orthogonal section?

3) In the section 4.5 the authors say that they have clear evidence for "subgrains and lattice distortions", but this is not evident in the figures. For instance the misorientation angle histogram in the Figs. 8a and 8b do not show a low angle misorientation peak as one would expect when quartz is deformed in the crystal plasticity field. I guess this is

related to the cut off misorientation angle chosen for the grain calculations in MTEX, so the authors have to provide new histograms where these peaks are more clear. This is also necessary because the EBSD map in the supplementary material does not show abundant subgrain boundaries.

4) A very interesting feature in these misorientation histograms is the lack of a misorientation peak at 60°, related to Dauphine twinning. Do the authors "cleaned" the twinning or the lack of twins is a real feature in this sample. If the later is the case, this should be discussed in the paper, as this is not very common in quartz EBSD data;

Some minor comments include:

Page 2, line 3-4 – I would briefly discuss these three different models like in two sentences each, that allows a quick comparison between models;

Page 3, line 4 – substitute "We present a highresolution map of porosity distribution on the mm scale in an ultramylonite and" by "Through this work flow we demonstrate...";

Line 16 – is there any temperature estimation for the deformation?

Line 20 – remove "?"

Page 4, equation 1 (and the others) – is there any reference for these equations, like Heilbronner's book?

Page 5, line 6 – please add where the EBSD data was acquired (Bern?);

Line 10 – Mainprice et al. is not the correct reference for MTEX, the correct is Hielscher & Schaeben 2008 - A novel pole figure inversion method: specification of the MTEX algorithm, J. of Appl. Cryst., 41(6), 2008.

Line 11 – please specify the parameters for the ODF calculations (halfwidth, etc). This is given in the figures but should be included here

Line 23 – what do you mean by thin section wafer?

Line 23 – "ion beam techniques";

Page 6, line 12 – please refer to the figures;

Line 20 – how calcic is the plagioclase? Please give an estimative of An content

Page 7, line 6 – please briefly explain how the hexbin statistic works;

Line 15 – how do you define high and low beta angles?

4.2.4 – the numerical definitions in the figure are different from the ones presented in the text

Line 17 – the authors mentioned 3 clusters, but I only see one, maybe the authors should indicate them in the figures;

Page 9. Line 6 – Fig. 6c

Line 8 – the authors mentioned that the dentrites are Si-rich. Looking at their Fig. 6, it is clear that the dentrites have a maximum of few 100's of nanometers in thickness. If the EDS was done with 20 kV as mentioned in the paper, the volume of interaction in quartz would be around 1 $\mu$m, meaning that the Si X-ray signal the authors detected may come from the quartz underneath. Did the authors performed low kV EDS analyses for better spatial resolution (less interaction volume)?

Page 10, line 6 – remove the two sentences between brackets, no need to call the questions here

Page 11, line 30 – dominant slip. . .plane? Direction?

Page 12, line 8 – how do you do EBSD on cavities? âŸž

Line 26 – remove "systematic"

Line 13, line 9-10 – in the way is written, it reads as if the fluids could induce pinning

Figures

General comment – either rotate all the pictures to have foliation E-W compatible with the pole figures in Fig. 8 (and the standard tectonic reference frame with foliation/lineation E-W) or rotate the pole figures from figure 8 to have the foliation N-S

Figure 1 – you should separate A, and C from the big picture B, and also write B on the big picture. You should also consider making A and C bigger, in the printed version the features are really small (consider also increasing the font size)

Figure 2 & 3 – I would make all the pictures bigger

Figure 4 – The authors mentioned in the caption that the largest pores have high beta values (with long axes parallel to X) but this is not clear from the Fig. 4b, there are only 7 or 9 points with blue colors (indicating larger pores) with high beta values, is this number relevant, considering that for intermediate size pores (yellow) you have much more points covering a full range of beta values?

Figure 5 – please make the scale for A & C and B & D the same, for easier comparison

Figure 6 – in the picture D you point to "incipient precipitates" (I imagine you are referring to the brig tiny spots), but the tip of your arrow points to an artefact caused by the grains from the coating, you should move the arrow to point exactly one of the bright spots or make a circle around the whole area

Figure 7 – Is there a colorcode for the 3D model of porosity? And is it possible to have the same orientation as X-Y-Z as in the 2D figures?

Figure 8 – your contoured pole figures are missing the primitive circle of the stereonet. The arrows pointing to quartz grain dispersion and quartz domain width should be more separate and the font larger. The font needs to be larger in the pole figure legend and in the histograms. Pleas also write [0001] instead of (0001).

Luiz F. G. Morales

---

## Author Comment (AC1) · 30 Oct 2017

Dear Editor,

We kindly thank the reviewers for their comments and will address each reviewer's comments separately. Where orthographic errors or changes in sentence structure have been suggested, we have changed spelling and, in some cases, structure. Therefore to keep this reply clutter-free we will only address those reviewer comments that pertain to the scientific content.

Firstly, we shall address the review of Jacques Précigout, presenting first the reviewer's comment and then the corresponding reply. Thereafter, we will address the comments of Luiz F.G. Morales.

Please find a copy of the revised manuscript below the authors replies with the pertinent changes highlighted.

Best,

James Gilgannon

Reviewer 1: Jacques Précigout

COMMENTS ON TEXT:

*Page 1-abstract: The object and techniques used have to be summarised in the abstract. In its present form, we have no idea of what the authors did.*

We have expanded the abstract to more specifically state the objective and techniques used.

*P3-L27: More information about the thicknesses of carbon coating have to be given here. Actually, in an ideal case, it is not recommended to use carbon coating for EBSD analyses, but sometimes, it is better to add a few nanometers to avoid ≪charging≫ effect. So what thickness did you put on your sample block ?*

We have added the thickness of the carbon and gold coatings to the manuscript.

*P5-L10: Please add the reference of Bachmann et al. (2010) for MTEX*

We appreciate the reviewer's direction on this matter but our citation choice for MTEX was made based on the guidance provided on the MTEX website (http://mtex-toolbox.github.io/publications.html). Here it is stated, "...please cite one of the following paper that best fits your application.". From the selection provided we chose Mainprice *et al.* (2011) because of its focus on deformation mechanisms and rheology. As the authors of MTEX have not specified a specific citation requirement we will retain our choice but thank the reviewer for his suggestion.

*P5-L11: Some precisions are here required concerning the minimum number of indexed points per grain. Is that in one row, several rows or for the whole grain? We commonly use a minimum threshold of 5 consecutive pixels in several rows.*

In our case it was a filtering condition of 10 indexed points per whole grain. We have amended the text accordingly.

*P9-L1: The figure 6d is called before figure 6b. Please, make sure that the figures are properly called in the manuscript (in successive order).*

We have corrected this mistake and now call figures in successive order.

*P10-L23: The figure 9a is not called anywhere.*

Figure 9 a is now called in the text (Note that figure 9 is now figure 10 in the manuscript).

*P11-L20: Please add the reference of Kassner and Hayes (2003) after ≪dislocations≫.*

We are aware of the work of Kassner and Hayes (2003) but feel that the inclusion of this work does not add to the literature already cited. Through out the manuscript we cite directly literature that the Kassner and Hayes (2003) review paper draws on. This choice was deliberate to provide a direct link to specifically relevant

papers for each statement. We provide the relevant citations after the next, more specific, sentence.

*P13-L29: It would be interesting to discuss our paper here (Précigout et al., 2017, Nature communications), which deals with the relationships between creep cavitation and phase nucleation in ultramylonites. The authors also have to discuss the recent paper of Cross and Skemer (2017, JGR solid Earth). Moreover, I have a question about your sentence claiming that fluid-filled cavities remain unfilled: if phase nucleation does not arise from creep cavitation, how do you explain the transition from GSI to GSS creep by phase mixing? In our paper in 2016 (Précigout and Stünitz, 2016), we do not claim that phase nucleation necessarily follows cavitation. We just say that phase nucleation is fast enough to maintain grain size small and randomize the olivine fabric. Some cavities may remain unfilled, particularly if the fluid is undersaturated.*

The reviewer raises several points with in his comment and we will address each point separately.

- *It would be interesting to discuss our paper here (Précigout et al., 2017, Nature communications), which deals with the relationships between creep cavitation and phase nucleation in ultramylonites.*

  We thank the reviewer for pointing us towards his contribution. At the time of writing our manuscript the first author had not read the work of Précigout *et al.* (2017) and so it was not included in the discussion. Précigout *et al.* (2017) presents compelling results, primarily from olivine slip systems and hydrous phase distributions, as evidence for fluid circulation in mantle rocks. While the results of this study are of interest to the topic of strain localisation and phase mixing at large, Précigout *et al.* (2017) do not present results that pertain to the processes of creep cavitation. Précigout *et al.* (2017) assume that cavities become active in their model but provide little direct evidence of creep cavities, or of a time evolution of phase mixing. Therefore while we appreciate the reviewer's suggestion we think that the discussion of the results of our study does not necessitate the inclusion of a discussion of Précigout *et al.* (2017). We have, however, amended the introduction to include a citation of Précigout *et al.* (2017) as a work that invokes creep cavities in their study.

- *The authors also have to discuss the recent paper of Cross and Skemer (2017, JGR solid Earth).*

  The reason for the exclusion of the paper from the discussion is that we wished to address the questions around the process of creep cavities explicitly, and we feel that the contribution of Cross and Skemer (2017) does not address this but targets more broad continuum mechanical questions of rheological weakening. While we agree that their contribution is a valuable one to the topic of localisation and rheology, their discussion focuses heavily on what they call "geometric phase mixing" which in simpler terms is the development of banding or compositional stratification. In terms of rheology, they discuss how this "geometric phase mixing" can contribute to mechanical weakening. However, the formation of compositional bands is not phase mixing *sensu stricto*. Their discussion focuses on what can be considered as a form of higher order clustering, something like Thomas clustering (Wiegand and Moloney, 2013). Our work focuses on the microstructures akin to the tail end of their experiments, the dispersal of monomineralic domains, or the development of anti-clustering. These results in the work of Cross and Skemer (2017) are given little attention in their discussion. Hence we feel that the work of Cross and Skemer (2017) does not present results that warrant discussion in the context of our results.

- *Moreover, I have a question about your sentence claiming that fluid-filled cavities remain unfilled: if phase nucleation does not arise from creep cavitation, how do you explain the transition from GSI to GSS creep by phase mixing? In our paper in 2016 (Précigout and Stünitz, 2016), we do not claim that phase nucleation necessarily follows cavitation. We just say that phase nucleation is fast enough to maintain grain size small and randomize the olivine fabric. Some cavities may remain unfilled, particularly if the fluid is undersaturated.*

  The question of phase mixing does not need to be restricted to two solid phases but can be extended to a solid-fluid system. Our rational behind this statement is as follows:

  1. If, as we speculate, the Zener-Stroh mechanism arises out of a creeping monomineralic quartz domain to provide sites of dilation, then there is a now a way to introduce a distributed second, fluid, phase into a single phase domain.

  2. In addition to this, fluid contributes to the continuum mechanical properties of the rock. A fluid filled pore would achieve this at the grain scale by acting to inhibit grain boundary migration and grain growth.

3. Furthermore, cavities that open will act as sites of low stress, encouraging the infiltration of a fluid. This can lower the adhesion and cohesion of grain boundaries, locally enhancing grain boundary sliding.

4. The final consequence of introducing creep cavities is that, even in the absence of solid phase precipitates, is that they perturb a homogenous domain into a two-phase rheological system.

COMMENTS ON FIGURES:

*Figure 1: Giving the GPS point is not enough, the authors have to provide a simplified map of Australia that locates the sample area. I would also recommend to add a picture of the outcrop. The figure 1C is not located. The figure 1B is not labelled.*

Our contribution focuses on understanding the evolution of processes active at the micro scale and therefore we feel that adding a map and field photo does not enhance the manuscript or that these form a necessary step in following the main arguments. We have provided several relevant references for the Redback Shear Zone if the reader wishes to gain further insight to the sample's field context.

*Figure 2 (caption): I think it is ≪figure 2e and f≫, not ≪figure 2e and d≫.*

We have changed the caption accordingly.

*Figures 6 and 7: these two figures arrive too late in the manuscript. They should appear after figure 1, particularly to show the microstructural features of pores. The text will have to be changed accordingly. By the way, the figure 1C has to be shown with figure 6. The figure 7 also demonstrates that the authors documents 3D features coeval with rock deformation. It has to be given before going into details concerning the distribution and shape of micro-cavities.*

The reviewer raises several points with in his comment and we will address each point separately.

- *Figures 6 and 7: these two figures arrive too late in the manuscript. They should appear after figure 1, particularly to show the microstructural features of pores. The text will have to be changed accordingly.*

We thank the reviewer for this suggestion, however, after much consideration we feel that we disagree. While we see the value in presenting the conventional microstructural images together and early, in the manuscript we frame the contribution around key quantitative results that drive the conceptual model. In this sense the observations in figure 6 on the broken surface are qualitative and supportive of the quantitative image analysis in the XZ plane. Furthermore due to the difficulties in processing the nCT data set no quantitative insights, with reasonable uncertainties, could be presented other than the connectivity. Therefore figure 7 is also mostly supportive of the observations in the XZ plane. We do concede that the results could be rearranged but we do not see any greater benefit to restructuring the manuscript so significantly.

- *By the way, the figure 1C has to be shown with figure 6.*

In line with our response above, we feel we disagree with the addition of figure 1c to figure 6, primarily because the figure is constructed with reference to observations in one plane of finite strain. We do not wish to confuse the reader by mixing planes of observation within a figure. However, we have added a clearer cross reference between figure 6b and 1c to draw the readers eye to similarities of pore shapes observed.

- *The figure 7 also demonstrates that the authors documents 3D features coeval with rock deformation. It has to be given before going into details concerning the distribution and shape of micro-cavities.*

We do not agree that Figure 7 alone documents the porosities coeval activity with rock deformation. On its own figure 7 shows that the porosity is pervasive and connects in 3D. It is the complete and contextualized set of observations presented in the paper that provides insight into when the pores were most likely active. In fact the two strongest lines of evidence are the change in pore shape congruent with the changes in microstructure and the corresponding changes in grain misorientation. Therefore we would argue, as above, that figure 7 is mostly supportive and there is not an obvious benefit to moving the figure.

*Figure 6 (caption): please details the sub-figures (a, b, c, etc.). I am not sure that the figure 6e is necessary.*

Some details of sub-figures have been added to the caption.

*Figure 8: The EBSD maps have to be shown in the manuscript (not in supplementary material), at least to show the sub-grains. I would recommend to show the three of them. Furthermore, the c axes have to be spelled between square brackets (≪ [0001] ≫) and the <a> axes are commonly indicated using <11-20>. The pole figure texture is based on, but not the ODF. Please change ≪ ODF ≫ by ≪ texture ≫ (or equivalent). Please provide the Mindex, as well. That will definitely confirm your point about the distribution of misorientation angles. Use ≪ uniform ≫ instead of ≪ random ≫ in the figure legend.*

The reviewer raises several points, we will address them separately:

- *The EBSD maps have to be shown in the manuscript (not in supplementary material), at least to show the sub-grains. I would recommend to show the three of them.*

  As advised we have moved the EBSD maps into the manuscript proper.

- *Furthermore, the c axes have to be spelled between square brackets (≪ [0001] ≫) and the <a> axes are commonly indicated using <11-20>.*

  We have changed the c-axis to be indicated with square brackets and we have changed the <a> family to the convention indicated by the reviewer.

- *The pole figure texture is based on, but not the ODF. Please change ≪ ODF ≫ by ≪ texture ≫ (or equivalent).*

  We have retained the label of ODF because it is the ODF that is displayed. We calculate the ODF and then display it in a pole figure. Therefore it is the ODF calculation that we are in fact visualising. We do not wish to mislead the reader into thinking that it is contoured point data we present.

- *Please provide the Mindex, as well. That will definitely confirm your point about the distribution of misorientation angles.*

  We see no need to present the M-index as we already present misorientation angle histograms for each subset. Furthermore the relative change in the J-index with the changing microstructure already validates, with an independent method, the changes seen between the misorientation angle histograms. Presenting the M-index calculation is therefore unnecessary.

- *Use ≪ uniform ≫ instead of ≪ random ≫ in the figure legend.*

  We have changed random to uniform in the figure legend.

*Line 11 – please specify the parameters for the ODF calculations (halfwidth, etc). This is given in the figures but should be included here*

We have added the parameters for the ODF calculations.

*Line 23 – what do you mean by thin section wafer?*

We were referring a thin-section and have removed the word 'wafer'.

*Line 20 – how calcic is the plagioclase? Please give an estimative of An content*

We thank the reviewer for asking because we mistakenly suggested the plagioclase was more calcic. In fact the plagioclase has an An content of $\sim 28\%$, making the plagioclase much more sodic than calcic. We have changed the text to reflect this.

*Page 7, line 6 – please briefly explain how the hexbin statistic works;*
*Line 15 – how do you define high and low beta angles?*

We have added a brief explanation of hexbin plots to the methods (3.3.1 Image analysis). An explanation of the definition of beta is given in the methods too (3.3.1 Image analysis: Pore orientation).

*4.2.4 – the numerical definitions in the figure are different from the ones presented in the text*

We have amended this discrepancy.

*Line 17 – the authors mentioned 3 clusters, but I only see one, maybe the authors should indicate them in the figures;*

As we do not wish to crowd the figure with extra annotations, we point to the clusters by coordinate position of the figure in the caption.

*Line 8 – the authors mentioned that the dentrites are Si-rich. Looking at their Fig. 6, it is clear that the dentrites have a maximum of few 100's of nanometers in thickness. If the EDS was done with 20 kV as mentioned in the paper, the volume of interaction in quartz would be around 1 μm, meaning that the Si X-ray signal the authors detected may come from the quartz underneath. Did the authors performed low kV EDS analyses for better spatial resolution (less interaction volume)?*

Unfortunately, we did not. However, we compensated for the larger interaction volume by increasing the aperture size and lengthening the count time. Reducing the interaction volume might possibly yield slightly different results but we consider this unlikely. Reducing the interaction volume would just change the ratio of counts between the quartz below and the surface itself. We feel that by counting for longer and thereby increasing the ability to catch more x-rays with an increased aperture compensated enough for a rough characterisation.

*Page 11, line 30 – dominant slip. . .plane? Direction?*

We have added direction to the text.

*Line 13, line 9-10 – in the way is written, it reads as if the fluids could induce pinning*

This is how we meant it to sound. Fluids are known to inhibit grain boundary mobility (see discussion of Reviewer 1's comments above). In our case, as in the case with a solid second phase, the dragging energy will compete with the surface energy to slow or inhibit grain boundary migration.

COMMENTS ON FIGURES:

General comment – either rotate all the pictures to have foliation E-W compatible with the pole figures in Fig. 8 (and the standard tectonic reference frame with foliation/lineation E-W) or rotate the pole figures from figure 8 to have the foliation N-S

We have rotated the figures to the standard tectonic reference frame with foliation/lineation E-W and have changed the text accordingly.

Figure 1 – you should separate A, and C from the big picture B, and also write B on the big picture. You should also consider making A and C bigger, in the printed version the features are really small (consider also increasing the font size)

We have revised figure 1 to better highlight the features.

Figure 2 & 3 – I would make all the pictures bigger

The figure sizes here reflect the Latex formatting. We have designed the figures so that they should be legible on a portrait A4 page.

Figure 4 – The authors mentioned in the caption that the largest pores have high beta values (with long axes parallel to X) but this is not clear from the Fig. 4b, there are only 7 or 9 points with blue colors (indicating larger pores) with high beta values, is this number relevant, considering that for intermediate size pores (yellow) you have much more points covering a full range of beta values?

We agree that the largest pores with the closest alignment to X are debatably relevant. They are potentially cracks that run parallel to the foliation and it is difficult to determine their genetic origin (i.e. post or syn-kinematic). However from figures 3b and c we know that there seems to be a continuum, defined by a power law relationship, between the intermediate pores and the largest pores. With this in mind suggest that it is hard to rule out the relevance of the small population of large pores aligned to X, primarily because the very nature of a power law relation dictates that there will be a small number of larger pores. Most of our line of discussion later does not focus on this small population of larger pores but we have added a note in the text to highlight the point that the reviewer raised.

Figure 5 – please make the scale for A & C and B & D the same, for easier comparison

We have set the scales to the same values and provided additional kernel density plots to verify the changes we claim occur with true point density analysis. We have added additional information in both the methods

and supplementary material sections to describe the parameters used in the kernel density analysis.

Figure 6 – in the picture D you point to "incipient precipitates" (I imagine you are refer- ring to the brig tiny spots), but the tip of your arrow points to an artefact caused by the grains from the coating, you should move the arrow to point exactly one of the bright spots or make a circle around the whole area

We have implemented the change suggested by the reviewer.

Figure 7 – Is there a colorcode for the 3D model of porosity? And is it possible to have the same orientation as X-Y-Z as in the 2D figures?

We have clarified the colour coding in the figure caption. Unfortunately we only know the sample's reference to the foliation plane. Therefore as we cannot decompose the foliation plane into X and Y, we cannot accurately provide a reference to the principle axes of finite strain. We have instead given reference in the caption to the figure's relationship to the foliation plane.

Figure 8 – your contoured pole figures are missing the primitive circle of the stereonet. The arrows pointing to quartz grain dispersion and quartz domain width should be more separate and the font larger. The font needs to be larger in the pole figure legend and in the histograms. Pleas also write [0001] instead of (0001).

We have implemented the changes requested by the reviewer.

[revised manuscript text omitted]